

# Slower maximal walking speed is associated with poorer global cognitive function among older adults residing in China

Guiping Jiang[1,2] and Xueping Wu[1]

[1] Shanghai University of Sport, School of Physical Education and Sport Training, Shanghai, China
[2] Harbin University, School of Physical Education, Harbin, Heilongjiang, China

## ABSTRACT

**Background**. Maintaining both walking speed and cognitive function is essential for active, healthy aging. This study investigated age-related differences in walking speed and global cognitive function with aging and the association between them among older adults residing in the developing country of China.

**Methods**. This cross-sectional study measured usual (UWS) and maximal walking speed (MWS) of participants for six meters. The Chinese version of the Montreal Cognitive Assessment was used to evaluate global cognition through in-person interviews. Analyses of variance were used to compare the differences in UWS, MWS, and global cognition between genders and age groups. Multiple linear regression models were used to determine the association between walking speed and global cognitive function.

**Results**. In total, 791 Chinese adults (252 men and 539 women) aged 60–89 years were included in this study. Markedly slowed UWS and worse global cognitive function scores were observed for both genders among adults ≥80 years of age. MWS slowed considerably in men ≥85 years of age and in women ≥80 years of age. There was a significant gender difference in MWS—with men walking faster than women—but not in UWS. Linear regression analysis adjusted for the confounding factors of gender, height, weight, years of education, and chronic disease indicated that MWS, but not UWS, was significantly associated with global cognitive function ($\beta = 0.086$, [0.177, 1.657], $P = 0.015$) such that slower maximal walking speed was associated with poorer cognitive performance. This association was statistically significant only for adults aged 75-79 years ($\beta = 0.261$ [0.647, 4.592], $P = 0.010$).

**Conclusion**. Walking speed was slower in the older age groups. Global cognitive function deteriorated markedly after 80 years of age. After controlling for confounding variables, slower MWS, but not UWS, was associated with poorer global cognitive function. MWS may serve as a potential indicator for earlier identification of poor cognition and motoric cognitive risk syndrome in an older Chinese population.

Corresponding author
Xueping Wu, wuxueping@sus.edu.cn

## INTRODUCTION

The global population is aging, and aging is often accompanied by impaired physical and cognitive functions (*Sofi et al., 2011*; *Clouston et al., 2013*) that may lead to decreased abilities to perform activities of daily living. As the most populous developing country in the world, China is poised to have a moderately aged population. Thus, there is an urgent need to study cost-effective and efficient indicators of physical and cognitive decline in older adults in China as well as in other developing countries. Walking, the most basic activity of daily living and an important determinant of the quality of life in later years, requires the coordination of multiple systems. Walking speed is considered the sixth vital sign (*Fritz & Lusardi, 2009*), after respiration, heartbeat, blood pressure, body temperature, and pain, and is a core indicator of health and functional ability in aging and disease (*Montero-Odasso et al., 2019*; *Stenholm et al., 2019*; *Rosso et al., 2013*; *Verghese et al., 2013*; *Verghese et al., 2019*). A slower walking speed may reflect a damaged system, a high energy cost of walking, or diminished motor control (*Studenski et al., 2011*). Therefore, maintenance of a normal and steady ability to walk for older adults is important for the prevention of adverse events in later life. Both usual walking speed (UWS) and maximal walking speed (MWS) have been used to predict frailty, falls, and mobility impairment in older adults (*White et al., 2013*).

Walking speed has been associated with cognitive function, with cognitive function referring to the process of acquiring or applying knowledge or to information processing (*Hunt, 1989*), and is the most basic human mental process. Safe and effective walking requires input from higher cognition areas (*Hausdorff et al., 2005*). Significant reduction in cognitive processing abilities have been shown among people who walk slowly, suggesting that walking speed may serve as a simple, noninvasive biomarker for early identification of poor cognition- (*Demnitz et al., 2016*; *Peel et al., 2019*; *Hirono et al., 2021*). Maintaining walking speed and cognitive function is essential for preventing motoric cognitive risk syndrome, and walking speed may be useful for identifying poor cognition.

Previous studies have investigated walking speed (*Hirono et al., 2021*) and global cognitive function with respect to age (*Boyle et al., 2021*) and gender differences (*Callisaya et al., 2008*). Other studies (*Fitzpatrick et al., 2007*; *Hao et al., 2021*; *Deshpande et al., 2009*; *Garcia-Pinillos et al., 2016*) have evaluated the association between walking speed and cognitive function. However, those studies mostly focused on developed countries and used different measurement methods. For instance, walking speed in the study by *Fitzpatrick et al. (2007)* was obtained using a 15-foot timed walking procedure, and cognitive function was measured by the Modified Mini-Mental State Examination (MMSE). While *Fitzpatrick et al. (2007)* and *Garcia-Pinillos et al. (2016)* both used the MMSE to assess cognitive function, Garcia-Pinillos and colleagues (*Garcia-Pinillos et al., 2016*) used a 10-metre rather than 15-foot (~4.57 metre) walk test. In addition, factors such as geographical differences may have affected the results of those studies (*Cai et al., 2020*). Systematic reviews and meta-analyses (*Demnitz et al., 2016*; *Peel et al., 2019*) indicate that most previous studies assessed UWS, only a few examined both UWS and MWS, and fewer still investigated the association of MWS with cognitive function. Those few studies have shown that MWS,

which is more physically challenging than UWS (*Sheridan et al., 2003*), is a better predictor than UWS of poor cognition in people with limited cognitive reserve (*Fitzpatrick et al., 2007*).

To date, research on age-related differences in walking speed and cognitive function and their association in older Chinese adults is scarce. Whether UWS, MWS or both are associated with cognitive function in this population has not been studied. Therefore, the present study aimed to investigate age-related differences in walking speed and global cognitive function and the association between them in older adults residing in the developing country of China. We hypothesized that older age groups would have slower walking speed and poorer cognitive function than younger age groups. We also hypothesized that MWS, but not UWS, would be significantly associated with global cognitive function.

## MATERIALS & METHODS

### Participants

The study population was drawn from older adults in eight communities in Shanghai. The inclusion criteria were as follows: (1) community-dwelling adult $\geq$60 years of age; (2) walk independently without the use of a walking aid; (3) have sufficient communication skills to complete the study; (4) agreeing to participate in this study. The exclusion criteria included the following:(1) an inability to understand the test; (2) a diagnosis of osteoarthritis, Parkinson's disease, dementia, stroke or a neurological disorder; and (3) declining to participate in the study. This study was approved by the Ethics Committee of Shanghai University of Sport (No. 102772021RT067). All participants provided written informed consent.

### Assessment of walking speed

The 6-meter walk is a common method for assessing walking speed (*Aoyagi et al., 2001*). Participants walked 6 m without assistance. Colored marking tape was applied to level ground at the starting position as well as at 2, 8, and 10 m. Before the test, the investigator explained and demonstrated UWS (habitual walking speed) and MWS (walking as fast as possible but not running). After hearing the word "start", the participant walked from the starting position to the marker at 10-m. All participants completed the test twice at their UWS and then completed the test once more at their MWS. The time needed to walk the middle 6 m was recorded to avoid the influence on the pace of the starting acceleration in the first 2 m and the braking deceleration in the last 2 m. Times were measured with a stopwatch, as *Peters, Fritz & Krotish (2013)* showed that a handheld stopwatch is as reliable as an automatic timer for measuring walking speed. The averaged time for each of the two tests performed at UWS was recorded and was considered accurate to 0.01 s. The final walking speed was calculated by dividing 6 m by the time required to complete the test. Walking speed was accurate to 0.01 m s$^{-1}$.

### Assessment of cognitive function

The Chinese version of the Montreal Cognitive Assessment (MoCA-C) was used to evaluate global cognition through in-person interviews. The MoCA-C was evaluated by uniformly

trained psychology researchers. The scale consisted of a total of 30 points: visual space and executive function (five points); attention (six points); delayed recall, memory (five points); naming (three points); language (three points); abstract reasoning (two points); and orientation (six points). The MoCA has been shown to be a reliable tool with high sensitivity and specificity for assessing cognitive function (*Nasreddine et al., 2005*).

## Confounders

Participants were invited to participate in face-to-face interviews to complete a questionnaire that asked about their age (Early 60s: 60–64 yrs, Late 60s: 65–69 yrs; Early 70s: 70–74 yrs, Late 70s: 75–79 yrs; Early 80s: 80–84 yrs, Late 80s: 85–89 yrs; respectively), gender, weight, height, and medical history, which included a history or physician diagnosis of hypertension, diabetes, hyperlipidemia, and heart disease. Those variables were considered confounders.

## Statistical analysis

Continuous variables are presented herein as the mean ± standard deviation, and non-normally distributed continuous variables, such as MoCA-C scores, are expressed herein as medians and quartiles. Baseline UWS, MWS, and MoCA-C scores as well as demographic characteristics were analyzed by independent-samples $t$-tests, Pearson's chi-square tests, or Mann–Whitney tests. Walking speed satisfied the condition of ANOVA (compliance with normal distribution and chi-square test). Two-way ANOVA was performed to compare the differences in walking speed between age groups of both gender for the presence of the main effects of age and gender. One-way ANOVA was used to compare the differences in walking speed between the age groups of different genders. A Kruskal-Wallis one-way analysis of variance was used to compare the differences in global cognition between the age groups. The Mann–Whitney test was used to compare the differences in variables between genders.

The results of our statistical tests indicated that there was no multicollinearity for the independent variables (variance inflation factors < 5) and that the residuals were normally distributed, indicating that the conditions for using linear regression were met. Global cognitive function was used as the dependent variable, and UWS and MWS were the independent variables. Multiple linear regression models were used to determine the association between walking speed and global cognitive function for all participants and for the different age groups. The main confounders included age, gender, weight, height, years of education, hypertension, diabetes, hyperlipemia, and heart disease. All statistical analyses were performed using SPSS, version 26.0, and $P < 0.05$ was considered statistically significant.

# RESULTS

## Participant characteristics

This cross-sectional study included 791 Chinese adults (252 men and 539 women) aged 60–89 years. Their characteristics are given in Table 1. No statistically significant differences between genders were detected for either UWS or for MoCA-C total scores, but there was

**Table 1  Characteristics of the study population.**

| Characteristic | Total | | Men | | Women | | *P* value |
|---|---|---|---|---|---|---|---|
| | *n* = 791 | | *n* = 252 | | *n* = 539 | | |
| Age, (years)[a] | 70.40 | 6.95 | 71.79 | 7.33 | 69.75 | 6.68 | <0.001** |
| Early 60s (n, %)[d] | 174 | 22.0 | 47 | 18.7 | 127 | 23.6 | |
| Late 60s (n, %)[d] | 246 | 31.1 | 67 | 26.6 | 179 | 33.2 | |
| Early 70s (n, %)[d] | 172 | 21.7 | 59 | 23.4 | 113 | 21.0 | |
| Late 70s (n, %)[d] | 95 | 12.0 | 31 | 12.3 | 64 | 11.9 | |
| Early 80s (n, %)[d] | 70 | 8.8 | 31 | 12.3 | 39 | 7.2 | |
| Late 80s (n, %)[d] | 34 | 4.3 | 17 | 6.7 | 17 | 3.2 | |
| Height (m)[c,a] | 1.61 | 0.08 | 1.69 | 0.06 | 1.57 | 0.06 | <0.001** |
| Weight(kg)[c,a] | 62.43 | 10.24 | 69.36 | 9.49 | 59.19 | 8.89 | <0.001** |
| BMI (kg m$^{-2}$) [c,a] | 24.08 | 3.28 | 24.37 | 2.98 | 23.95 | 3.40 | 0.072 |
| ≥12 years of education[d] (n, %) | 499 | 63.08 | 174 | 69.05 | 325 | 60.30 | <0.001** |
| Walking speed (m s$^{-1}$)[a] | | | | | | | |
| UWS[c] | 1.22 | 0.26 | 1.25 | 0.02 | 1.21 | 0.01 | 0.057 |
| MWS[c] | 1.62 | 0.36 | 1.70 | 0.02 | 1.58 | 0.01 | <0.001** |
| History of disease (n, %) | | | | | | | |
| Hypertension[d] | 395 | 49.9 | 130 | 51.6 | 265 | 49.2 | 0.526 |
| Diabetes[d] | 155 | 19.6 | 51 | 20.2 | 104 | 19.3 | 0.756 |
| Hyperlipemia[d] | 147 | 18.6 | 32 | 12.7 | 115 | 21.3 | 0.004** |
| Heart disease[d] | 200 | 25.3 | 60 | 23.8 | 140 | 26.0 | 0.514 |
| MoCA-C[e,b] | 26 | (24–28) | 26 | (24–28) | 26 | (24–28) | 0.580 |

Notes.

Early 60s represents ages between 60 and 64 years; late 60s represents ages between 65 and 69 years; early and late years are similarly separated for the 70s and 80s age groups.

BMI, body mass index; UWS, usual walking speed; MWS, maximal walking speed; MoCA-C, Chinese version of Montreal Cognitive Assessment.

[a]Values are expressed as mean ± standard deviation.

[b]Values are expressed as median and quartiles.

[c]Independent-samples *t*-test.

[d]Chi-square test.

[e]Mann–Whitney test.

*$P < 0.05$

**$P < 0.01$

a significant gender difference in MWS, with men walking significantly faster than women.

## Age-related differences in walking speed

The analysis of Two-way ANOVA revealed no significant interaction between UWS and MWS. However, there were significant main effects of age and gender for UWS and MWS. These results suggested that both UWS and MWS slowed with increasing age. Post-hoc tests assessing age groups of both genders showed that both UWS and MWS for people in their late 80s were significantly slower than for those in the other age groups (Table 2). For both genders, UWS and MWS in the early 80s age group were significantly slower than UWS and MWS in the early or late 60s age groups and in the early 70s age group (Figs. 1 and 2). UWS and MWS in the late 60s and 70s age groups were significantly slower than UWS and MWS in the early 60s age group. UWS in the early 70s age group was significantly slower than UWS in the early 60s age group. MWS for men in the early 80s age group was

**Table 2   Age-related differences in walking speeds and cognitive performance in men and women.**

| | Early 60s N = 174/47/127 | Late 60s N = 246/67/179 | Early 70s N = 172/59/113 | Late 70s N = 95/31/64 | Early 80s N = 70/31/39 | Late 80s N = 34/17/17 |
|---|---|---|---|---|---|---|
| **Walking speeds**, mean (standard deviation) | | | | | | |
| UWS(m s⁻¹) | | | | | | |
| Both genders | 1.33(0.24) | 1.25(0.24)* | 1.23(0.24)** | 1.17(0.27)** | 1.11(0.27)**††§ | 0.85(0.22)**††§§♦♦××× |
| Men | 1.33(0.23) | 1.28(0.25) | 1.20(0.22) | 1.23(0.28) | 1.21(0.28)▲ | 0.89(0.25)**††§§♦♦×× |
| Women | 1.33(0.24) | 1.23(0.24)* | 1.24(0.24) | 1.15(0.27)** | 1.04(0.30)**††§§ | 0.82(0.17)**††§§♦ |
| MWS(m s⁻¹) | | | | | | |
| Both genders | 1.76(0.32) | 1.64(0.31)** | 1.65(0.33) | 1.54(0.38)** | 1.42(0.35)**††§§ | 1.16(0.33)**††§§♦♦××× |
| Men | 1.85(0.35)▲ | 1.71(0.34)▲ | 1.66(0.35) | 1.64(0.39) | 1.59(0.29)*▲ | 1.25(0.39)*†§♦× |
| Women | 1.72(0.30) | 1.62(0.29) | 1.65(0.32) | 1.50(0.37)** | 1.28(0.34)**††§§♦ | 1.06(0.23)**††§§♦♦ |
| **MoCA-C** total score, median (quartiles) | | | | | | |
| Both genders | 27 (25, 28) | 27 (25, 28) | 26 (25, 28) | 26 (24, 28) | 23 (20, 26)**††§§♦♦ | 23 (17, 26)**††§§♦♦ |
| Men | 27 (25, 28) | 27 (25, 28) | 26 (25, 28) | 27 (26, 29) | 24(21, 27)*††♦ | 24 (17, 26.5)*††♦ |
| Women | 27 (24, 29) | 27 (25, 28) | 26 (25, 29) | 26 (23, 28) | 21(19, 26) **††§§♦ | 23 (17, 25)**††§§ |

Notes.

MoCA-C, Chinese version of Montreal Cognitive Assessment.

Early 60s represents ages between 60 and 64 years; late 60s represents ages between 65 and 69 years; early and late years are similarly separated for the 70s and 80s age groups.

*Significant difference compared with the early 60s group (* $P < 0.05$, ** $P < 0.01$).

† Significant difference compared with the late 60s group (†† $P < 0.01$).

§ Significant difference compared with the early 70s group (§§ $P < 0.01$).

♦ Significant difference compared with the late 70s group (♦ $P < 0.05$, ♦♦ $P < 0.01$).

× Significant difference compared with the early 80s group (× $P < 0.05$, ×× $P < 0.01$).

▲ Significant difference compared with women (▲ $P < 0.05$, ▲▲ $P < 0.01$).

significantly slower than MWS for men in the early 60s age group. For women, UWS and MWS in the late 80s age group were significantly slower than UWS and MWS in the other age groups. UWS and MWS in the early 80s age group were significantly slower than UWS and MWS in the early and late 60s and in the early 70s age groups. UWS and MWS in the late 70s age group were significantly slower than UWS and MWS in the early 60s age group. Moreover, MWS in the early 80s group was significantly slower than MWS in the late 70s group. Post-hoc independent t-tests to compare the differences in MWS between genders showed significant differences between genders in the early and late 60s age groups. There were significant differences in UWS and MWS between genders in the early 80s age group.

## Age-related differences in global cognitive function

This cross-sectional study found that global cognitive function among older adults in this cohort was significantly lower in people ≥80 years of age (Table 2). The results of Mann–Whitney tests assessing age groups among both genders showed that for men, global cognitive functioning scores in the early 80s age groups were significantly lower than those scores in the other age groups. For women, global cognitive function scores in the early and late 80s age groups were significantly lower than those scores in the early and late 60s age groups and in the early 70s age groups. Additionally, global cognitive function scores in the early 80s age group were significantly lower than those scores in the late 70s group. Gender differences in global cognitive function were not statistically significant.

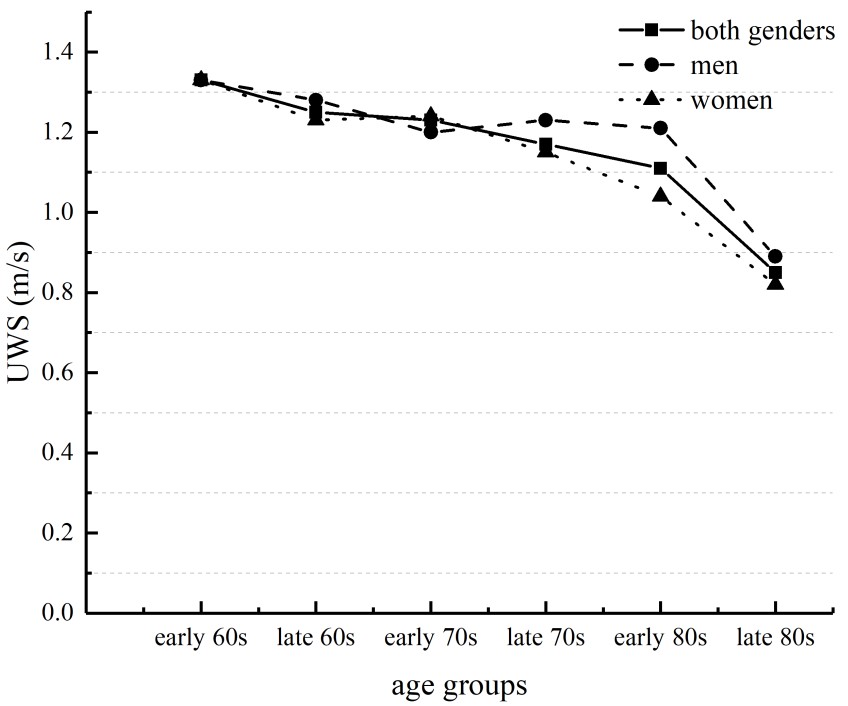

**Figure 1 Age-related differences in usual walking speed (UWS) in men and women.** Early 60s represents ages between 60 and 64 years; late 60s represents ages between 65 and 69 years; early and late years are similarly separated for the 70s and 80s age groups.

## Associations between walking speed and global cognitive function

Overall, the results of multiple linear regressions indicated no significant association between UWS and global cognitive function ($P > 0.05$) (Table 3). By contrast, MWS was significantly associated with global cognitive function in Model 1 (adjusted for gender, age, height, and weight), in Model 2 (adjusted for gender, age, height, weight and years of education) and in Model 3 (adjusted for gender, age, height, weight, years of education, hypertension, diabetes, hyperlipemia, and heart disease) ($\beta = 0.086$, [0.177, 1.657], $P = 0.015$). These results suggested that faster MWS was associated with higher global cognitive function. Further analysis by age groups revealed that MWS and global cognitive function were significantly correlated only in the late 70s age group ($\beta = 0.261$, [0.647, 4.592], $P = 0.010$).

## DISCUSSION

Our findings supported our hypothesis that both UWS and MWS slowed with age in adults ≥60 years of age. Markedly slower UWS and MWS were observed for both genders among people ≥80 years of age. Compared with those in other groups, UWS and MWS were significantly slower after the late 80s for men and after the early 80s for women. Our results showing that walking speed slowed with age were consistent with those found worldwide (*Tolea et al., 2010*; *Bohannon, 1997*; *Busch et al., 2015*). However, the UWS of the older adults in our study population was significantly faster than that of older people

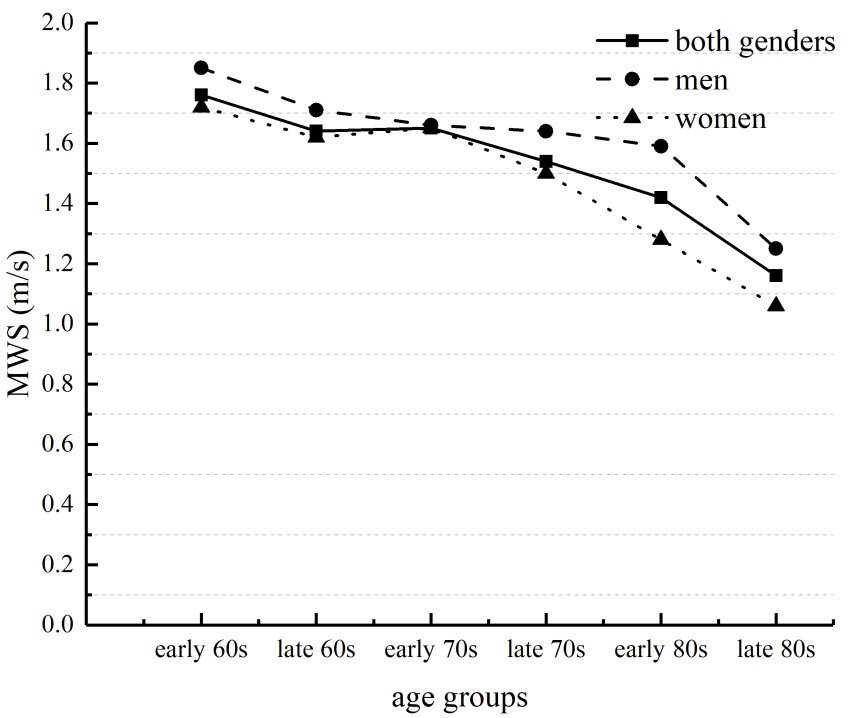

**Figure 2** **Age-related differences in maximal walking speed (MWS) in men and women.** Early 60s represents ages between 60 and 64 years; late 60s represents ages between 65 and 69 years; early and late years are similarly separated for the 70s and 80s age groups.

in some other countries (1.08 m s$^{-1}$) (*Cai et al., 2020*) but slower than that among older Japanese adults (men, 1.39 m s$^{-1}$; women, 1.31 m s$^{-1}$) (*Tanimoto et al., 2012*), despite the adults in the latter study being older than our cohort. That study by Tanimoto and colleagues included community-dwelling participants aged ≥65 years whose health status was unknown and assessed UWS using at a distance of 5-m. The participants in our study were community-dwelling adults aged ≥60 years with known chronic disease status (Table 1) and UWS was assessed at a distance of 6-m. Thus, factors such as the baseline health of the study participants may affect the results. In addition, UWS may vary by the population studied and the methodology used to assess it (*Busch et al., 2015*). For example, Busch and colleagues' study (*Busch et al., 2015*), the effects of start-up acceleration and braking deceleration on walking speed were not considered, whereas in our study the above-mentioned factors that interfere with walking speed measurements were excluded. Previous studies have shown that MWS among older Japanese adults is markedly slower after the age of 70 years (*Peters, Fritz & Krotish, 2013*). In the present study, compared with the participants in the early 60s age group, the MWS of men was significantly slower in the early 80s age group, and the MWS of women in the late 70s groups was significantly slower. In addition, there were gender differences in MWS, mainly in the early and late 60s age groups and in the early 80s age group. A previous study (*Hunt, 1989*) has shown markedly slower walking speeds in women than in men, consistent with our study. The walking

**Table 3 Cross-sectional associations between walking speed and global cognitive function in the population.**

| Walking speed (m s$^{-1}$) | Model 1 | | | | Model 2 | | | | Model 3 | | | |
|---|---|---|---|---|---|---|---|---|---|---|---|---|
| | β | Wald 95% Confidence limits | | P | β | Wald 95% Confidence limits | | P | β | Wald 95% Confidence limits | | P |
| **UWS** | | | | | | | | | | | | |
| Total | −0.019 | −0.752 | 1.292 | 0.605 | 0.007 | −0.879 | 1.070 | 0.848 | 0.006 | −0.890 | 1.069 | 0.858 |
| Early 60s (n = 174) | −0.130 | −3.543 | 0.287 | 0.095 | −0.120 | −3.381 | 0.374 | 0.116 | −0.107 | −3.231 | 0.554 | 0.164 |
| Late 60s (n = 246) | −0.065 | −2.491 | 0.819 | 0.321 | −0.084 | −2.688 | 0.524 | 0.187 | −0.093 | −2.818 | 0.445 | 0.153 |
| Early 70s (n = 172) | 0.096 | −0.855 | 3.633 | 0.224 | 0.092 | −0.890 | 3.544 | 0.239 | 0.078 | −1.151 | 3.396 | 0.331 |
| Late 70s (n = 95) | 0.042 | −2.323 | 3.496 | 0.690 | 0.088 | −1.597 | 4.078 | 0.387 | 0.122 | −1.217 | 4.656 | 0.248 |
| Early 80s (n = 70) | −0.048 | −4.869 | 3.270 | 0.696 | −0.069 | −4.558 | 2.265 | 0.504 | −0.061 | −4.481 | 2.450 | 0.560 |
| Late 80s (n = 34) | 0.149 | −5.866 | 13.900 | 0.412 | 0.073 | −7.492 | 11.411 | 0.674 | 0.100 | −7.339 | 12.691 | 0.586 |
| **MWS** | | | | | | | | | | | | |
| Total | 0.109 | 0.392 | 1.937 | 0.003** | 0.088 | 0.204 | 1.681 | 0.012* | 0.086 | 0.177 | 1.657 | 0.015* |
| Early 60s (n = 174) | −0.047 | −1.870 | 1.007 | 0.554 | −0.041 | −1.791 | 1.025 | 0.592 | −0.023 | −1.640 | 1.207 | 0.765 |
| Late 60s (n = 246) | 0.027 | 0.690 | −1.062 | 1.603 | 0.006 | 0.142 | 0.373 | 0.929 | 0.004 | −1.264 | 1.353 | 0.947 |
| Early 70s (n = 172) | 0.152 | −0.009 | 3.098 | 0.051 | 0.1440 | −0.121 | 2.959 | 0.071 | 0.126 | −0.315 | 2.877 | 0.115 |
| Late 70s (n = 95) | 0.203 | 0.025 | 4.057 | 0.047* | 0.223 | 0.303 | 4.180 | 0.024* | 0.261 | 0.647 | 4.592 | 0.010* |
| Early 80s (n = 70) | −0.017 | −4.096 | 3.600 | 0.898 | −0.108 | −4.776 | 1.711 | 0.349 | −0.104 | −4.721 | 1.772 | 0.367 |
| Late 80s (n = 34) | 0.110 | −4.723 | 8.550 | 0.560 | 0.076 | −4.879 | 7.525 | 0.665 | 0.105 | −4.703 | 8.374 | 0.567 |

Notes.

UWS, usual walking speed; MWS, maximal walking speed.

Model 1 adjusted for age, gender, weight, height; Model 2 adjusted for age, gender, weight, height, and years of education.

Model 3 adjusted for age, gender, weight, height, years of education, hypertension, diabetes, hyperlipemia, and heart disease.

*$P < 0.05$

**$P < 0.01$

speed of women in the present study was slower than that of men. One study (*Guadagnin et al., 2019*) showed that changes in walking speed were strongly associated with the aging process and that this association is most significant in older women. Walking speed in men was predicted by brain white matter hyperintensity volume rather than by the degree of brain atrophy or magnetization transfer ratio peak height (adjusted for age and brain size). However, in women, slower walking speed was associated with lower magnetization transfer ratio peak height (suggestive of microstructure cerebral changes), increased white matter hyperintensity, and greater brain atrophy (*Rosano et al., 2010*).

There were no significant differences in global cognitive function among older adults for the groups encompassing 60 to 79 years of age. However, global cognitive function was poorer for participants ≥80 years of age. This result is in line with our hypothesis. The trend for lower global cognitive function in older adults was essentially the same for both genders, consistent with the results of a previous study (*Chinese Cooperative Group of Guidelines for Diagnosis and Treatment of Dementia and Cognitive Impairment, 2018*). The present study showed that global cognitive function remained stable until 80 years of age, which is consistent with previous studies (*Boyle et al., 2021*).

After adjusting for confounders in the present study, only MWS, not UWS, was significantly associated with global cognitive function in adults 60–89 years of age, which is also in line with our hypothesis. Notably, this association was statistically significant in the late 70s age group. Years of education and chronic diseases may affect walking speed and cognition in older adults. One study showed that compared with UWS, MWS was a more sensitive indicator of neuromuscular function (*Annweiler et al., 2010*). A previous longitudinal study of older Italian adults (*Deshpande et al., 2009*) and a cross-sectional study of older Japanese adults (*Fitzpatrick et al., 2007*) both showed that MWS was more associated with cognitive function than UWS. In addition, some studies (*Deshpande et al., 2009*) have shown that MWS is a better predictor than UWS of cognition. Postural control decreases with age. In addition to the involvement of the sensory system and the musculoskeletal system during postural control, cognitive function is critical for postural stability. The higher demands placed on the balance control system at MWS necessitate much higher conscious control and cortical activity in older adults than is required for usual walking (*Deshpande et al., 2009*). Thus, the ability to maintain good performance during rapid walking may be closely related to the integrity of cortical function, which is associated with good cognitive performance (*Deshpande et al., 2009*). The associations between walking speed and executive function, memory, and processing speed have been summarized in the literature (*Demnitz et al., 2016*). Numerous mechanisms may underlie slower walking speed in older people. For example, magnetic resonance imaging has shown that slower walking speed in older adults is associated with an increased proportion of subcortical white matter hyperintensities and periventricular (*Murray et al., 2010*) and hippocampal atrophy (*Callisaya et al., 2013*). However, executive function, a major domain of cognitive function, is also influenced by white matter hyperintensities. That is, the association between walking speed and cognitive function may be based in part on the involvement of common neural networks (*Murray et al., 2010*).

The results of the present study suggested that MWS was significantly associated with poorer global cognitive function in older adults, particularly for people in the late 70s age group. That may be related to the results of the present study showing a significantly slower MWS in the late 70s age group in women and a smaller sample size in women aged ≥80 years and in all men. It was extremely important to further understand this association, particularly in people over the age of 75 (*Fitzpatrick et al., 2007*). Our finding suggests that MWS may be used as a potential indicator for early identification of poor cognition or motoric cognitive risk syndrome. Furthermore, our in-depth analysis of the association between different walking speeds and global cognitive function aims to provide a simple,
easy to use, and sensitive indicator for clinical and practical assessments, and to provide a basis for proactive responses to cognitive impairment in older adults residing in developing countries. This study also identified numerous confounding factors associated with the association between walking speed and global cognition that are modifiable, including weight, hypertension, hyperlipidemia, and diabetes. Community health managers may prevent or delay declines in walking speed and cognitive function in older adults by helping them to regulate some of these modifiable factors. The results of this study also reinforce the clinicians' perception of walking speed as a sixth vital sign. Early recognition of motoric cognitive risk syndrome (*Verghese et al., 2014*), a pre-dementia syndrome characterized by both walking speed slowing and cognitive concerns, has led to an increased interest in preventing and delaying the deterioration of walking speed and cognition. This would provide a valuable approach to health management for healthy aging in developing countries, such as China.

The strengths of the present study were that we assessed the differences in walking speed and global cognitive function with age divided into 5-year intervals among adults aged 60–89 years and residing in China, a developing country. Thus, this study assessed both wide and narrow age ranges. MWS was found to be significantly associated with global cognitive function among older adults, especially those in the late 70s age group, in the Chinese community. Our results provide a reference for other relevant studies, especially in developing countries. This study also has limitations. Because this was a cross-sectional study, we could not explore the causal relationship between walking speed and global cognition. The small sample of women participants in the late 80s age group and all men participants may have biased the interpretation of the association. We studied only the associations between walking speeds and global cognitive function. Future studies should be conducted to analyze the associations of UWS and of MWS with subdomains of cognitive function. In addition to the assessed factors affecting the association between walking speed and cognition in this study, other factors may affect UWS, MWS, and global cognitive function. Thus, future studies should consider increasing the sample size of oldest adults, controlling for additional confounding factors, and assessing the associations of UWS and of MWS with cognitive function subdomains using broad neuropsychological test batteries in longitudinal studies.

## CONCLUSIONS

The results of this cross-sectional study indicated that both UWS and MWS slowed with age. The slowing of walking speed was most pronounced in the oldest age groups assessed. The present study also showed that global cognitive function remained stable until 80 years of age but deteriorated markedly after that. There were gender differences for UWS and MWS, but not for global cognitive function, among older adults. After controlling for gender, age, height, weight, years of education and common chronic diseases, we found that MWS was significantly associated with global cognitive function, whereas UWS was not. These results suggest that MWS may serve as a potential indicator for earlier identification of poor cognition and motoric cognitive risk syndrome in an older Chinese population.

## ACKNOWLEDGEMENTS

We are grateful to Shangti Health Technology (Shanghai) Co., Ltd. for their support in recruiting participants and to Prof. Wu's research group for their assistance in data collection.

### Funding

This study was supported by The Program for Overseas High-Level Talents at Shanghai Institutions of Higher Learning (No. TP2020063), the Heilongjiang Province Key Commissioning Project (SJGZ20200098), and the Ministry of Education Humanities and Social Sciences Program (21YJC890053). The funders had no role in study design, data collection and analysis, decision to publish, or preparation of the manuscript.

### Grant Disclosures

The following grant information was disclosed by the authors:
The Program for Overseas High-Level Talents at Shanghai Institutions of Higher Learning: TP2020063.
Heilongjiang Province Key Commissioning Project: SJGZ20200098.
Ministry of Education Humanities and Social Sciences Program: 21YJC890053.

### Competing Interests

The authors declare there are no competing interests.

### Author Contributions

- Guiping Jiang conceived and designed the experiments, performed the experiments, analyzed the data, prepared figures and/or tables, authored or reviewed drafts of the article, and approved the final draft.
- Xueping Wu conceived and designed the experiments, authored or reviewed drafts of the article, and approved the final draft.

### Human Ethics

The following information was supplied relating to ethical approvals (i.e., approving body and any reference numbers):

This study was approved by the Ethics Committee of Shanghai University of Sport (No. 102772021RT067).

### Data Availability

The measurement data are available in the Supplemental File.

### Supplemental Information

Supplemental information for this article can be found online at http://dx.doi.org/10.7717/peerj.13809#supplemental-information.

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
