# Peer review of "Slower maximal walking speed is associated with poorer global cognitive function among older adults residing in China"

_PeerJ, doi:10.7717/peerj.13809_

## Round 0.1 · original submission · Major Revisions

The three reviewers and I see considerable merit in the manuscript you have submitted. However, they also have identified a number of ways in which the manuscript can be improved.

·

Basic reporting

a. Lines 64-72: There’s a lot of good information in this paragraph, but I’m having trouble identifying the main argument. For example, what is the “cognitive clock,” and has previous work supported the idea that physical function could be a proxy measure of the cognitive clock?
b. I was surprised to see how much discussion there was on the prevalence of MCI in the discussion section (Lines 228-250). Was one of the main goals of this manuscript to describe the prevalence of MCI among older adults? If not, it’s not clear to me how this paragraph informs the results beyond demonstrating the generalizability of the results.

Experimental design

a. Were participants excluded based on any cognitive criteria?
b. How many walking speed trials were done per condition (i.e., number of usual gait speed trials and fast pace trials)? What were participants told to do in each condition? Were the gait trials done in the same order for all participants, or were the conditions counterbalanced?
c. “Confounders” subheading-> It is mentioned in Table 1’s note what the age categories represent. I would recommend also including this information in the manuscript under this heading.
d. Line 153: UWS and MWS [differed] by age. Since you indicate the directionality of the results throughout the rest, it may be good to do that here (even if it’s in the expected direction).

Validity of the findings

a. Lines 212-213: It’s noted that UWS may not be consistent across different ethnogeographic regions as well as one’s health status. Compared to other cohorts, is the prevalence of health conditions different among this sample? It is also possible that variability in UWS could be driven by the sample age; were the articles cited different in the sample age? It appears as though the older adults in the Tanimoto et al. article were somewhat older but walked faster.
b. What sort of conclusions should we draw, if any, regarding UWS versus MWS as a measure of central nervous system integrity and function?
c. Lines 251-273: This paragraph does a nice job summarizing why MWS may be more related to cognition than UWS. Do the authors anticipate that this relationship would hold across different cognitive domains? For example, it may be that MWS and UWS are equally predictive of memory, but MWS may be more related to speed of processing/attention than UWS since there is a speeded component that is more apparent in MWS (or something).

Additional comments

a. What is the correlation between usual and rapid gait speed?
b. Because these analyses rely on cross-sectional data, I would recommend referring to them as, e.g., age-related differences rather than changes.
c. Was the association between gait and cognition similar across the different age bands? For example, the association between UWS and cognition may be stronger among adults in their late 80s compared to those in their early 60s. This could answer whether there are ages where UWS and MWS predict cognition to a similar degree (e.g., maybe in the terminal decline stage).

·

Basic reporting

1. Basic Reporting
a. Clear and unambiguous, professional English used throughout.
The language is mostly ok. Terminology commonly used in longitudinal studies needs revising. Occasionally sentences are missing the comparator e.g. MWS was significantly slower after the early 80s for men and after the late 70s for women – is this compared to all other age groups, the youngest or the one directly preceding?
b. Literature references, sufficient field background/context provided.
I would suggest that reviews, since there are several available, should be used in the introduction.
c. Professional article structure, figures, tables. Raw data shared.
The article is appropriately structured, with figures and tables.
The figures would benefit from including a measure of spread/variability. The figures y-axis do not start at zero, but there is no axis break to indicate this.
Raw data has been shared. There is missing data that has not been addressed in the study.
d. Self-contained with relevant results to hypotheses.
There are many research questions presented in the results (age-related gait decline, age-related cognitive decline, each of these are presented by sex), but really only one clearly articulated in the introduction (relationship between gait and cognitive decline). No hypothesis is provided.

Experimental design

2. Experimental design
a. Original primary research within Aims and Scope of the journal.
Original research within scope of the journal.
b. Research question well defined, relevant & meaningful. It is stated how research fills an identified knowledge gap.
The primary aim is articulated. I feel the novelty and evidence gap that is being addressed needs to be more clearly articulated since there is already much literature in this area.
c. Rigorous investigation performed to a high technical & ethical standard.
The study received ethical approval. The model provided in Table 3 examining the relationship between cognitive performance and gait speed has been adjusted for potential confounders. Additional factors that have not been reported could affect gait speed.
d. Methods described with sufficient detail & information to replicate.
The methods are mostly clear, I have provided comment on aspects that could be improved in the PDF.

Validity of the findings

3. Validity of the findings
a. Impact and novelty not assessed. Meaningful replication encouraged where rationale & benefit to literature is clearly stated.
The relationship between gait and cognition is not novel. The study could better articulate the need for this study in the introduction.
b. All underlying data have been provided; they are robust, statistically sound, & controlled.
Measure of spread/variance for gait speed by age category figure (or report values in table) would benefit the reader. There is missing data that has not been reported.
The between group comparisons have not been adjusted for confounders, however, the main aim has been adjusted. There are some other factors that may influence gait speed/cognition e.g. dementia and OA.
c. Conclusions are well stated, linked to original research question & limited to supporting results.
The abstract conclusion goes beyond what is reported in the abstract results. The study conclusion is in line with the study findings.

Additional comments

Summary
This cross-sectional study has examined gait speed (usual pace and maximal) and cognitive function in 791 Chinese older people. The study has shown that gait speed and cognitive function is reduced in older age groups and that cognitive function, measured by the Chinese MoCA, is related to maximal but not usual gait speed. Sex differences in walking speed and cognitive decline were also investigated. There was no sex difference in cognitive performance. Maximal but not usual walking speed was affected by sex.
Main concerns:
1. My main concern is the lack of novelty of the manuscript. The evidence gap that this research is addressing needs to be better highlighted/presented more convincingly in the introduction. There is much literature on the association between cognitive function and gait, what does this study add that hasn’t been done already. Is it that it is in Chinese older people, if so, why is important to understand this relationship in this group of people? Is it the use of two walking speeds (this has also been examined before: Callisaya 2017, Umegaki 2018), but is less common? Is it the sex contrasts (this has been done as per authors)? What is the study adding?
2. Some of the group sizes are quite small secondary to the subgroups, particularly in the oldest group when examined by sex. The number of people with MCI in these groups, although a high proportion are still a very small number of people - which can reduce generalisability of the findings. Additionally, there are a lot of comparisons that involve quite small subgroup sample sizes (gait, MoCA, MCI prevalence).
3. For all but Table 3, the only statistical details provided are p-values and in some parts, not even this is provided (age-related gait changes).
4. The definition used for MCI was a cut-point on a cognitive scale, but did not include other diagnostic criteria for MCI. This needs to be discussed in more detail (as a limitation) particularly in relation to the higher prevalence of MCI in the current study. Also, I couldn’t find anything to indicate that people with dementia were excluded, which could also be contributing to the higher prevalence. If people with dementia were not excluded, this medical condition should be used as a confounder in the linear regression models.
5. The discussion compares the results to existing studies, but there is little discussion on what this study actually adds to the literature. Also, more on what the findings mean would be beneficial?

Additional comments:
I have provided a detailed review by annotating the PDF.
A couple of points for consideration:
It should be clear in the abstract that the walking distance is actually 6m (x2).
All of the tables are to 2 decimal places, which I feel makes the tables difficult to read and that this level of sensitivity is not required.
There is a lot of double reporting in tables/text – I will leave this to the editor to determine what is appropriate for the journal, however, unless it is a main finding I would suggest that the authors only need to report results once.
Age-related change in gait speed: There doesn’t seem to be any actual statistics reported for this section, just statements about significance.
The figures presenting gait speed lack standard deviations/confidence intervals. I have suggested adding raw data to a table, alternatively, SD or CI need to be added to the figures.
In the introduction, I feel that it would be better to use systematic reviews to support statements.
Line 79 is without a reference/s and I feel that it is not really accurate based on the evidence (a list is provided in the PDF).
Table 2 describes ‘Changes in total MoCA….’ in the title, but this is a cross-sectional study, so this needs to be revised.
Using the term ‘Decline’ can imply that the study is longitudinal. This should be revised so that it is clear that you are examining between group differences cross-sectionally. E.g. In the groups that were 80 years or older, MoCA scores were significantly lower than other age groups.
Please include sample sizes in the tables, so that they can standalone.
There are several typographical issues e.g. spacing.
The ethical review files are mislabelled. The English version is labelled as Chinese for one document and vice versa.

Suggestion: It may be interesting to look at the gain in gait speed from UWS to MWS and its relationship with cognition.

Reviewer 3 ·

Basic reporting

The introduction provides clear rationale, whilst methods and results are clearly presented, with a nice balance struck between text/table/figures, and the discussion provides context to the findings.

However, of importance for attention, there are some issues around presentation of data which I believe need to be addressed with this submission. Firstly, the formatting of Table 2 (to clarify significant differences) and Table 3 (to prevent runover onto multiple lines) needs some tidying up, to help reader understanding and clarify/simplify the many annotations within these tables.

Secondly, there is a need for a thorough proofread of this submission, to rectify the various minor spelling and grammatical errors and inconsistences present throughout the manuscript. Within this, there is a need to ensure a consistent reference approach is maintained throughout; there is an instance in the discussion where referencing slips to Numbered (line 219), which needs rectifying (and checking for other instances).

Experimental design

The research has deployed a valid experimental design; following sound methodological and statistical approaches and using established assessment tools and appropriate cut-offs. Methods and results are clearly presented, with a nice balance struck between text/table/figures. The authors go on to provide valid comparison to relevant research in the Discussion, providing context to findings and presenting plausible background to the relationship reported in this study (between MWS and cognitive function).

Validity of the findings

This research highlights the relationship between walking speed and cognitive function, framing this within the context of a developing country. As such, the findings can help inform the use of walking speed as a potential indicator of early cognitive decline; to that end, this research has good potential to help inform future research and clinical practice.

Additional comments

As above, this research highlights the relationship between walking speed and cognitive function, framing this within the context of a developing country. As such, the findings can help inform the use of walking speed as a potential indicator of early cognitive decline; to that end, this research has good potential to help inform future research and clinical practice.

I thank the authors for their presentation of a clear and robust investigation of the association between walking speed and cognition, discussing the ensuing relationship within the context of relevant research and appropriately acknowledging study limitations/making recommendations for future research.

---

## Round 0.2 · Minor Revisions

While the first reviewer is now happy with your revisions, the second reviewer still has some minor comments for you to address before this paper can be accepted for publication in PeerJ.

·

Basic reporting

No Comment

Experimental design

No Comment

Validity of the findings

No Comment

Additional comments

Thank you to the authors for your thoughtful incorporation of the reviewer and editor feedback. This is a well-conducted and written study, and I look forward to citing it in the future! The authors should be commended on a job well done.

·

Basic reporting

a. Clear and unambiguous, professional English used throughout.
The language is mostly ok. Terminology commonly used in longitudinal studies still needs revising.
b. Literature references, sufficient field background/context provided.
Improved from first submission
c. Professional article structure, figures, tables. Raw data shared.
The article is appropriately structured, with figures and tables.
Raw data has been shared. There is no longer any missing data.
d. Self-contained with relevant results to hypotheses.
Yes

Experimental design

a. Original primary research within Aims and Scope of the journal.
Original research within scope of the journal.
b. Research question well defined, relevant & meaningful. It is stated how research fills an identified knowledge gap.
The primary aim is articulated. The authors have improved on the rationale for the study.
c. Rigorous investigation performed to a high technical & ethical standard.
The study received ethical approval. The models provided in Table 3 examining the relationship between cognitive performance and gait speed have been adjusted for potential confounders. Additional factors that have not been reported could affect gait speed, but this has been commented on in the discussion. It is not clear how the confounders for Table 2 were selected.

Validity of the findings

a. Impact and novelty not assessed. Meaningful replication encouraged where rationale & benefit to literature is clearly stated.
Rationale for study is clear.
b. All underlying data have been provided; they are robust, statistically sound, & controlled.
Yes
c. Conclusions are well stated, linked to original research question & limited to supporting results.
The conclusions are in line with the study findings, though the relationship between cognition and gait speed was only significant in the late 70’s group.

Additional comments

Summary
The authors’ have revised the manuscript and have responded appropriately to many of my concerns. There still a few items that need to be considered.

Main concerns:
1. The small group sizes across the age groups for men needs to be acknowledged in the limitations section – this is not just in the older age group.
2. The authors have still used the term decline in several places e.g. Title, when this tends to indicate that a study is longitudinal. The authors have corrected this in some instances but not others (I have noted this in the attached annotated PDF).
3. There are still several instances where the sentences don’t make sense, or it is not clear what message is trying to be conveyed (I have noted this in the attached annotated PDF).
4. It is not clear how the covariates for the ‘Age-related differences in walking speed’ were established. Additionally, for MWS both genders and women, the authors have controlled for both height and BMI, but height contributes to BMI, so why not just control for height and weight?
5. The gender difference in walking speed may be attributable to height? Worth investigating?

Additional comments:
I have provided a detailed review by annotating the PDF.
There are several typographical issues e.g. spacing.
This suggestion was not considered: It may be interesting to look at the gain in gait speed from UWS to MWS and its relationship with cognition, but appreciate it may be out of the scope of the current manuscript

---

## Round 0.3 · Minor Revisions

I think the authors for attending to virtually all of the comments of the reviewers. However, a small number of very minor issues still need to be corrected before this paper is accepted for publication in PeerJ, including the following (with these line numbers representing the track changes word document):

Line 82: it should be rewritten as "methods. For instance,.....".
Line 84: it should be written as "Mini-Mental State Examination (3MSE). In contrast, walking speed was assessed by Garcia-Pinillos ....".
Line 299: it should be written as "For example, Busch and colleagues'...." and at this point provide the reference as I feel it's important you actually include the exact reference here and not just refer to their study in words.

---

## Round 0.4 · Minor Revisions

Thanks for attending to my suggestions. However, I have noticed some other small errors, including one typographical error that you made based on my initial suggestion. The requested revisions relate to the tracked changes document.

Line 71: provide the reference for Fitzpatrick and colleagues at the end of the sentence;

Line 73: replace 3MSE with MMSE;

Line 73-75: it should be rewritten as " While Fitzpatrick et al. and Garcia-Pinillos and colleagues (PROVIDE REFERENCE HERE) both used the MMSE to assess cognitive function, Garcia-Pinillos and colleagues (PROVIDE REFERENCE HERE) used a 10-metre rather than 15-foot (~4.57 metre) walk test.

---

## Round 0.5 · accepted · Accept

Thanks for attending to all of the requested changes.